# DeSKO: Stability-Assured Robust Control of Nonlinear Systems with a Deep Stochastic Koopman Operator

**Minghao Han**[1,2]**, Jacob Euler-Rolle**[2]**, Robert K. Katzschmann**[2]
1 Department of Control Science and Engineering, Harbin Institute of Technology
2 Soft Robotics Lab, ETH Zurich
`{minhan,ejacob,rkk}@ethz.ch`

## Abstract

The Koopman operator theory linearly describes nonlinear dynamical systems in a high-dimensional functional space and it allows to apply linear control methods to highly nonlinear systems. However, the Koopman operator does not account for any uncertainty in dynamical systems, causing it to perform poorly in real-world applications. Therefore, we propose a deep stochastic Koopman operator (DeSKO) model in a robust learning control framework to guarantee stability of nonlinear stochastic systems. The DeSKO model captures a dynamical system's uncertainty by inferring a distribution of observables. We use the inferred distribution to design a robust, stabilizing closed-loop controller for a dynamical system. Modeling and control experiments on several advanced control benchmarks show that our framework is more robust and scalable than state-of-the-art deep Koopman operators and reinforcement learning methods. Tested control benchmarks include a soft robotic arm, a legged robot, and a biological gene regulatory network. We also demonstrate that this robust control method resists previously unseen uncertainties, such as external disturbances, with a magnitude of up to five times the maximum control input. Our approach opens up new possibilities in learning control for high-dimensional nonlinear systems while robustly managing internal or external uncertainty.

## 1 Introduction

Modeling is crucial to the development of intelligent machines and robots. It is needed to predict the behavior of a dynamical system, analyze a system's interior properties, realize a reliable planning and control (Müller et al., 2007). For many systems of interest, such as a soft robotic arm that can interact with an unknown object (George Thuruthel et al., 2018), the governing equations are either unknown or too complex to be useful (Bakker et al., 2020). Thus, it is essential to use an accurate but concise model.

The need for a fast model that is able to represent complex systems has led to recent advancements in data-driven modeling of complex dynamics, and learning world models. These advancements have been achieved by constructing surrogate neural network (NN) models (Chua et al., 2018; Ha & Schmidhuber, 2018; Zhang et al., 2019). By formulating the modeling problem as a supervised learning task, the neural network can be trained to infer the systems' dynamics based on the dense state and action input (Nagabandi et al., 2018b) or high-dimensional image observations (Hafner et al., 2019). However, as nonlinear systems are often approximated by complex and over-parameterized neural networks (Morton et al., 2019), the learned models are restrictive when they are used for dynamical analysis and controller design. Optimal control problems based on NN models are typically non-convex, and they need to be solved by using inefficient population-based optimization methods (Bharadhwaj et al., 2020), such as random shooting (Zhang et al., 2019; Nagabandi et al., 2018a) or cross entropy methods (Chua et al., 2018; Hafner et al., 2019; Wang & Ba, 2019).

Recently, the Koopman operator theory has attracted much attention. The Koopman operator can model complex nonlinear systems by linearly propagating observables in an infinite dimensional functional space (Koopman, 1931b). This propagation allows us to apply the well-established linear control and analysis theory to nonlinear systems. Based on a system's runtime data, a linear operator, which propagates the observables in time, is then learned. Based on this idea, practical algorithms, like

dynamic mode decomposition (DMD) and extended dynamic mode decomposition (EDMD), were developed by selecting a finite set of parameterized linear and nonlinear observable functions. These methods have been applied to the modeling of soft robotic arms (Bruder et al., 2020; Haggerty et al., 2020), multirotors (Folkestad et al., 2020), and mobile robots (Shi & Karydis, 2021). More recently, Deep-DMD exploits deep learning techniques to automate the design of observable functions (Lusch et al., 2018; Otto & Rowley, 2019; Han et al., 2020; Yeung et al., 2018; Morton et al., 2018). Deep-DMD automatically learns a richer set of observable functions than DMD or EDMD, and it can also be more accurate, even though it uses fewer observables (Bakker et al., 2020). However, these methods focus on modeling nominal systems with clean data sets, while noise, either in the process or observation, has not yet been accounted for. Furthermore, it is challenging to ensure closed-loop stability in a controller design that is based on a learned Koopman operator model (Bakker et al., 2020).

In this work, we learn and control uncertain nonlinear dynamics using the deep stochastic Koopman operator (DeSKO) model. Instead of encoding the observables deterministically, DeSKO infers the probabilistic distribution of observables and learns a linear operator that can propagate the inferred distribution forward in time. Furthermore, we propose an efficient and robust model predictive controller based on the DeSKO model. In terms of robotic control benchmarks, we show that our proposed method outperforms state-of-the-art deep Koopman operators (Lusch et al., 2018; Han et al., 2020) in modeling and in stabilizing systems that have been corrupted by process and observation noises. Furthermore, we show that the proposed framework is scalable; it can handle high-dimensional and complex systems, including legged robots, soft robotic arms, and biological gene networks.

The outline of this paper is as follows. In Section 2, we introduce and discuss the related works. In Section 3, we introduce the basic notations and concepts of the Koopman operator (KO), and we present our proposed DeSKO model. In Section 4, we discuss our robust control framework and prove its stability. In Section 5, we evaluate the DeSKO model in terms of modeling, control, and robustness in eight different environments, and we compare it to baseline methods. Finally, in Section 6, we conclude the paper and discuss future challenges.

## 2 RELATED WORKS

**Koopman operators** Following the Koopman operator theory formalized in Koopman (1931a), various practical algorithms have been proposed to extract the Koopman spectral properties from state-transition data. DMD and EDMD adopt a hand-crafted set of linear and nonlinear functions to encode the observables and obtain the linear operator through least square optimization (Schmid, 2010; Tu et al., 2014; Williams et al., 2015). A straightforward approach to automate the design of observable functions is to parameterize the observable functions as neural networks and use deep learning techniques for training (Lusch et al., 2018; Otto & Rowley, 2019; Yeung et al., 2018; Morton et al., 2018; Takeishi et al., 2017). Unlike DMD and EDMD, Deep-DMD approaches do not suffer from the curse of dimensionality; that is, the number of observables does not grow exponentially with the number of states. Furthermore, Deep-DMD can also express a richer class of observable functions than DMD or EDMD. Graph neural networks and block-wise Koopman operators are introduced in (Li et al., 2020) to model objects composed of similar building blocks, such as ropes.

While the described works have greatly advanced the field of modeling and control by applying the Koopman operator, several challenges still remain. Azencot et al. (2020) proposed a Koopman autoencoder framework that can learn a Koopman model from noisy data for long horizon prediction, but only considered autonomous systems without control input. Morton et al. (2019) proposed to measure uncertainty with an ensemble of Koopman models sampled from the encoded distribution. However, their work is focused on deterministic systems with clean datasets. Furthermore, how to guarantee stability with a learned Koopman model remains an open question. Mamakoukas et al. (2020); Kolter & Manek (2019); Pan & Duraisamy (2020) studied how to learn a stable dynamics of embeddings on the latent space, while we are focused on stabilizing the original system with the learned model. Our method differs from previous works by learning a stochastic model from noisy data, and providing a robust control framework with a stability guarantee. We show that this framework can easily generalize to systems with process and observation noises.

**Model-based Learning Control** In recent years, model-based reinforcement learning has dominated a large portion of model-based learning control studies (Pascanu et al., 2017; Hamrick et al., 2017; Racanière et al., 2017). Several researchers have combined learned neural network models with model predictive control (Chua et al., 2018; Zhang et al., 2019; Hafner et al., 2019; Nagabandi et al., 2018a; Wang & Ba, 2019). In this work, we utilize the Koopman operator's linear propagation

property, and we conduct model predictive control efficiently. Furthermore, we also show that the Koopman operator substantially facilitates analyzing and assuring stability of the closed-loop system.

Gaussian process-based methods are another popular class of learning control methods (Kocijan et al., 2004; Berkenkamp et al., 2017; Vinogradska et al., 2016; Berkenkamp & Schoellig, 2015). As a type of non-parametric method, Gaussian process-based methods are efficient in modeling with small data sets. Indeed, Gaussian process models have been applied to various robotic systems (Chang et al., 2017; Cao et al., 2017; Kabzan et al., 2019; Chang et al., 2020). Nonetheless, Gaussian process-based approaches are typically limited to low-dimensional systems and small data sets (Wang et al., 2020). In this work, we present a framework that is scalable to high-dimensional robotic control tasks.

## 3 Modeling

In this section, we will first introduce the basic notations and concepts of the Koopman operator (KO) (Koopman, 1931a). Then the derivation of the DeSKO model is established.

### 3.1 The Koopman Operator

For the sake of clarity, we introduce the KO using notations that are similar to the ones in Bakker et al. (2020). Consider the nonlinear discrete time dynamical system

$$x_{t+1} = f(x_t, u_t)$$

where $f$ is a nonlinear differentiable function. The system can be lifted to an infinite-dimensional function space $\mathcal{F}$ composed of all square-integrable real-valued functions within the compact domain $\mathbb{X} \times \mathbb{U} \subset \mathbb{R}^{n+m}$. Elements of $\mathcal{F}$ are called *observables*. On this space, the flow of the systems is characterized by the Koopman operator $\mathcal{K} : \mathcal{F} \to \mathcal{F}$, which is an infinite-dimensional linear operator that satisfies

$$\mathcal{K}\psi(x_t, u_t) = \psi \circ f(x_t, u_t)$$

where $\psi$ denotes the observable function. However, the infinite-dimensional function space $\mathcal{F}$ and the KO $\mathcal{K}$ are impractical. Therefore, a finite dimensional function space $\overline{\mathcal{F}} \subset \mathcal{F}$, spanned by a linearly independent basis function $\psi : \mathbb{R}^n \to \mathbb{R}^h$ is used instead, where $h$ is the dimensionality of the observable function specified by the designer. Here, $i$ denotes the index of each observable function. The linear property of the KO enables the use of linear control techniques for efficient control. The Koopman equation can be written as

$$A\psi(x_t) + Bu_t = \psi \circ f(x_t, u_t) \tag{1}$$

where the KO is split into the Koopman matrix $A \in \mathbb{R}^{h \times h}$ and the control matrix $B \in \mathbb{R}^{h \times m}$, . For the reader's convenience, we also have also included the details for learning Koopman matrices with a set of observables in the Appendix.

The Koopman operator conducts a deterministic inference of observables and ignores the existence of any uncertainty that may stem from the system and the lack of data about the system. A controller design based on such a defective model would inevitably perform suboptimally, and even result in failure. In this work, we are interested in learning and controlling a nonlinear system that has been corrupted by additive noise:

$$x_{t+1} = f(x_t, u_t) + w_t \tag{2}$$

where $w_t \sim p_w$ denotes random noises subject to the distribution $p_w$. In this case, the transition of states is probabilistic, and it is denoted by $p(x_{t+1}|x_t, u_t)$. In the next section, we will discuss how to overcome this challenge by encoding and propagating the distribution of observables with Koopman operators.

### 3.2 Deep Stochastic Koopman Operator

The deep stochastic Koopman operator (DeSKO) consists of two building blocks: a probabilistic neural network that encodes the distribution of the observables, and a KO that propagates the distribution into the future. In the following section, we will describe the construction, implementation, and optimization of the DeSKO.

**Probabilistic neural network** To account for uncertainty, we exploit a probabilistic model $p_\theta(\psi_t|x_t)$ to encode the distribution of observables given the current state. The probabilistic model is parameterized by a probabilistic neural network. The output neurons of the probabilistic NN characterize

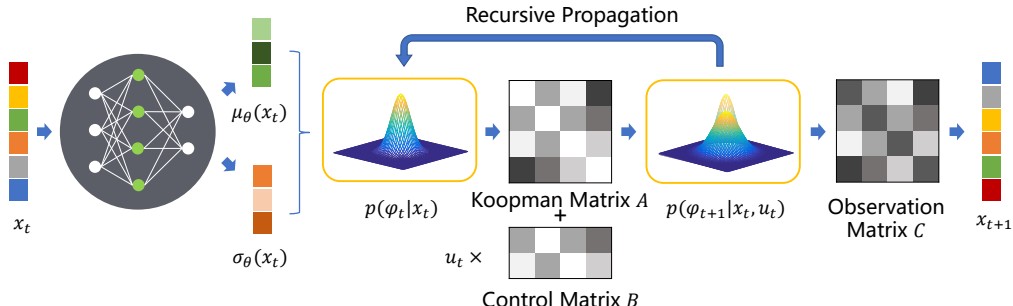

Figure 1: Structure of the Deep Stochastic Koopman Operator (DeSKO). With the help of two neural networks, the DeSKO model encodes the system state $x_t$ into the parameters of a Gaussian distribution of observables, which are the mean vector $\mu_\theta(x_t)$ and the variance vector $\sigma_\theta(x_t)$. Then, the learned Koopman matrix $A$ and control matrix $B$ propagate the distribution to the next time step. This can recursively produce a series of distribution predictions over a given time horizon. Finally, the learned observation matrix $C$ maps the observables to the state space.

the parameters of the distribution to capture the uncertainty caused by the process and observation noise. In our particular case, we design two NNs to output a mean vector $\mu_\theta(x_t)$ and a diagonal covariance vector $\sigma_\theta(x_t)$. Together, these vectors describe a multivariate Gaussian distribution, which is defined as $p_\theta(\psi_t|x_t) = \mathcal{N}(\mu_\theta(x_t), \sigma_\theta(x_t))$ (see also Figure 1). A Gaussian distribution is often chosen for continuous-valued states, and it is a reasonable choice if the uncertainty in the system is unimodal (Chua et al., 2018). While a Gaussian distribution is effective in our experiments, other tractable distributions can be used instead. In order to expressively encode the dynamics, the distribution parameters can be represented with an arbitrarily high complexity by using nonlinear functions that depend on the current state. This makes it feasible to incorporate NNs into the probabilistic model.

**Propagation of observable distributions** As shown in Figure 1, given a series of control input $\{u_t, u_{t+1}, \ldots, u_{t+T}\}$, the distribution of observables encoded by the probabilistic NNs is recursively propagated into the future with the Koopman matrix $A$ and control matrix $B$. According to (1), the KO propagates the observables pointwise on the functional space $\overline{\mathcal{F}}$. By conducting pointwise mapping, KO is also able to infer the future distribution of observables. The use of KO to conduct distribution propagation can also be justified from the perspective of the Frobenius-Perron Operator (FPO) and has been discussed in the Appendix. Finally, the decoder maps the observables back to the state space and produces the prediction results. The decoder is designed to be linear, and parameterized by an observation matrix $C \in \mathbb{R}^{n \times h}$. This facilitates controller design, which will be discussed in the next section.

**Optimization procedure** To optimize the probabilistic neural network and the Koopman operator, including the observation matrix, the single-step prediction loss is given as

$$\mathcal{L}_t(A, B, C, \theta) := \mathbb{E}_{p_\theta(\psi_t|x_t)} \|x_{t+1} - C(A\psi_t + Bu_t)\|$$

where $\mathbb{E}$ denotes expectation. However, the model trained with single-step optimization may not satisfactorily perform long-term predictions, since it suffers from the accumulation of prediction errors during recursive propagation. To this end, we train the model to minimize the following multi-step prediction loss:

$$\mathcal{L}(A, B, C, \theta) = \mathbb{E}_{\mathcal{D}} \sum_{k=1}^{H} \mathbb{E}_{\psi_{t+k}} \|x_{t+k} - C\psi_{t+k}\| \tag{3}$$

where $\mathcal{D}$ denotes the data set composed of multiple $T$ steps input trajectories $u_{0:T}$ and the resulting state trajectories $x_{0:T}$. In (3), $H$ denotes the horizon of forward prediction. The distribution of the observable $\psi_{t+k}$ is inferred through recursive propagation with the KO based on the current state $x_t$ and a series of control inputs $u_{t:t+k-1}$. We exploit the reparameterization trick (Haarnoja et al., 2018) in the calculation of (3) to achieve a more stable training process (Xu et al., 2019). By injecting a Gaussian noise vector $\epsilon_i \sim \mathcal{N}(0, I)$, the expectation of the observables can be approximated by $\frac{1}{N} \sum_{i=0}^{N} (\mu_\theta(x_t) + \epsilon_i \sigma_\theta(x_t))$. This could produce a lower variance estimator of the gradient, and also enables the straightforward exploitation of KO in forward propagation. The optimization problem is

summarized as

$$\min_{A,B,C,\theta} \mathcal{L}(A,B,C,\theta) = \mathbb{E}_\mathcal{D} \sum_{k=1}^{H} \sum_{i=0}^{N} \|x_{t+k} - C\psi_{t+k}^i\|$$

$$\text{s.t. } \psi_{t+k}^i = A\psi_{t+k-1}^i + Bu_{t+k-1}$$

$$\psi_t^i = \mu_\theta(x_t) + \epsilon_i \sigma_\theta(x_t), \ \epsilon_i \sim \mathcal{N}(0,I) \tag{4}$$

The parameters in (4) are updated using gradient descent.

**Entropy Constraint** To further improve the robustness of the DeSKO model, we introduce a constraint on the entropy of the observable distributions during optimization (4). During the training process of probabilistic models, the entropy of the learned distribution naturally falls as the model becomes more certain about the underlying dynamic. However, the lack of data in some states may cause the learned model to become overly confident about the encoding. To this end, we prevent the model from overfitting by constraining the average entropy of the learned distribution above a minimum entropy threshold, i.e., $\mathbb{E}_\mathcal{D} - \log p(\psi_t|x_t) \geq \mathcal{H}$, where $\mathcal{H}$ denotes the minimum entropy threshold. Note that the constraint is valid for the average entropy, rather than a pointwise constraint on the state space, thus the entropy can vary at different states. In our implementation, we exploit the Lagrange method to conduct the constrained optimization. The entropy constraint is multiplied by a Lagrange multiplier and added into the loss function. During the training of NNs, we also adjust the weight of the entropy constraint by updating the Lagrange multiplier using gradient ascent.

## 4 CONTROL

In this section, we will present the robust control framework for the proposed DeSKO model. We exploit the model predictive control (MPC) framework to achieve robust and stabilizing control guarantees. First, a nominal MPC controller (Borrelli et al., 2017) is designed to solve a finite horizon stochastic optimal control problem, minimizing the expectation of cumulative stage cost. As the DeSKO model encodes a Gaussian distribution of the observables, the nominal dynamic of the expectation of observables could be characterized by

$$\hat{\mu}_{t+1} = A\hat{\mu}_t + Bc_t \tag{5}$$

where $\hat{\mu}_t$ is the nominal mean vector encoded by the DeSKO model and $c_t$ denotes the nominal control input. Then the nominal MPC solves the following deterministic optimal control problem

$$V^*(\hat{\mu}_t) = \min_{c_{0:H-1}} \sum_{k=0}^{H-1} \|C\hat{\mu}_{t+k}\|_Q^2 + \|c_{t+k}\|_R^2 + \|C\hat{\mu}_{t+H}\|_P^2 \tag{6}$$

$$\text{s.t. } \hat{\mu}_{t+k+1} = A\hat{\mu}_{t+k} + Bc_{t+k}, c_{t+k} \in \mathbb{U} \tag{7}$$

where $Q$, $R$ and $P$ are known positive definite weight matrices, and $r$ denotes the reference signal.

Ideally, after applying an optimized control input $c_t^*$ to the system, the mean vector encoded at the next time step $\mu_\theta(x_{t+1})$ should equal to $A\hat{\mu}_t + Bc_t^*$. However, this can hardly be true due to the existence of noise. The actual dynamic of the mean vector is dominated by

$$\mu_{t+1} = \mu_\theta(f(x_t, u_t) + w_t) \tag{8}$$

With the learned KO, the evolution of the actual mean vector on the observable space is defined as

$$\mu_{t+1} = A\mu_t + Bu_t + g(w_t) \tag{9}$$

where $g(w_t) = \mu_\theta(f(x_t, u_t) + w_t) - \mu_\theta(f(x_t, u_t))$ characterizes the unknown effect of $w_t$ on the observable dynamics.

To compensate for the control error caused by the uncertainty, a stabilizing feedback controller $K$ is introduced to drive the actual state $\mu_t$ to the nominal trajectory $\hat{\mu}_t$. At every time step, the action input is given by

$$u_t = c_t^* + K(\mu_\theta(x_t) - \hat{\mu}_t) \tag{10}$$

In this work, the feedback controller $K$ is obtained by using the linear quadratic regulator (LQR).

To ensure the closed-loop stability of the nominal system (5), the following constraint needs to be satisfied (Borrelli et al., 2017; Mayne et al., 2000),

$$\|C(A+BK)\hat{\mu}_t\|_P^2 - \|C\hat{\mu}_t\|_P^2 \leq -\|C\hat{\mu}_t\|_Q^2 - \|K\hat{\mu}_t\|_R^2 \tag{11}$$

In fact, by exploiting the LQR controller $K$, this constraint can be easily satisfied. The matrix $P$ is determined by solving the following discrete time Riccati equation,

$$P = C^T Q C + A^T P A - A^T P B (R + B^T P B)^{-1} B^T P A \tag{12}$$

and the $K$ is given by

$$K = -(R + B^T P B)^{-1} B^T P A \tag{13}$$

The overview of the control procedure is summarized in Algorithm 1.

### 4.1 STABILITY WITH EXACT KOOPMAN OPERATORS

First, we investigate the case where the exact Koopman operators are available. To establish the stability guarantee, we make the following assumptions:

**Assumption 1.** *The probabilistic NN $\mu_\theta$ is Lipschitz continuous, $\|\mu_\theta(x + y) - \mu_\theta(x)\| \leq L\|y\|$.*

**Assumption 2.** *The random noise has bounded energy, i.e., a finite constant $b$ exists such that $\mathbb{E}_w \|w\| \leq b$.*

With the assumptions above, we can establish the stability guarantee as follows:

**Proposition 1.** *Consider system (2) controlled by the Robust MPC controller (6)-(7) and (10). Then the closed-loop system (2) is uniformly ultimately bounded with bound $\frac{\beta \sigma L b}{1 - \beta}$, where $\beta$ denotes the maximum eigenvalue of the closed-loop transition matrix $A_K = A + BK$ and $\sigma := \|C\|$.*

The detailed proof of the proposition is deferred to the Appendix.

### 4.2 STABILITY WITH APPROXIMATED KOOPMAN OPERATORS

Proposition 1 shows that the controller is stabilizing with the exact Koopman operator. However, in practice, the exact Koopman matrices $A^*, B^*, C^*$ are generally infeasible and only the sub-optimal solutions $A, B, C$ can be obtained. Now, we will show that stability could be assured even though the Koopman matrices are approximated with bounded prediction error.

First, the nominal system (5) is a conceptual system constructed with the approximated Koopman matrices and thus its dynamic and the resulting MPC problem remain the same as (5), (6) and (7). On the other hand, the evolution of the mean vector is characterized with the exact Koopman matrices,

$$\mu_{t+1} = A^* \mu_t + B^* u_t + g(w_t)$$
$$x_t = C^* \mu_t$$

We define the dynamic residual caused by the sub-optimal approximation $\epsilon_t := (A^* - A)\mu_t + (B^* - B)u_t$, and the reconstruction residual $d_t := (C^* - C)\mu_t$. Then the above system can be rewritten as

$$\mu_{t+1} = A\mu_t + Bu_t + g(w_t) + \epsilon_t$$
$$x_t = C\mu_t + d_t$$

**Assumption 3.** *There exist positive constants $\gamma, \eta \in \mathbb{R}^+$, such that $\|\epsilon\| \leq \gamma$ and $\|d\| \leq \eta$.*

With the Assumptions 1-3, we can establish the stability guarantee as follows:

**Proposition 2.** *Consider system (2) controlled by the Robust MPC controller (6)-(7) and (10). Then the closed-loop system (2) is uniformly ultimately bounded with bound $\frac{\beta \sigma (Lb + \gamma)}{1 - \beta} + \eta$, where $\sigma := \|C\|$.*

Detailed proof of the above proposition is referred to Appendix. Proposition 2 shows that with the controller given by (10), the stability of the system (2) can still be guaranteed even with approximated Koopman matrices, though the uniform ultimate bound is inevitably larger than the ideal case in Proposition 1.

**Remark 1.** *Assumption 3 is equivalent to assuming that $\|A^* - A\|$, $\|B^* - B\|$, and $\|C^* - C\|$ are bounded respectively, and the state space $\mathbb{X}$ and action space $\mathbb{U}$ are bounded. To simply the notations, we chose the form in Assumption 3.*

In addition to the stabilization tasks, we are also interested in how to achieve optimal tracking control with the DeSKO model. By plugging in a reference signal and modifying the input regulation accordingly, the proposed method could be extended to deal with set-point and dynamic tracking problems. The algorithmic and implementation details are detailed in the Appendix.

---

**Algorithm 1** Robust MPC with DeSKO

---

**Require:** Weighting matrices $Q, R, P$, state feedback matrix $K$, prediction horizon $H$
    **Initialize** $\hat{\mu}_1 \leftarrow \mu_\theta(x_1)$
    **for** $t = 1, 2, \ldots$ **do**
        Solve (6)-(7) to obtain $c_t^*$
        Apply $u_t = c_t^* + K(\mu_\theta(x_t) - \hat{\mu}_t)$ to the system (2)
        $\hat{\mu}_{t+1} \leftarrow A\hat{\mu}_t + Bc_t^*$
    **end for**

---

## 5 EXPERIMENTS

In this section, we will evaluate the performance of the DeSKO model in terms of modeling, control and robustness. Specifically, we evaluate the following aspects: (a) **Convergence** of the proposed training algorithm with random parameter initialization; (b) **Model performance** of the DeSKO compared to other baselines in learning and predicting diverse dynamics; (c) **Reliability** of the control framework achieving successful performance and stability guarantees; (d) **Robustness** of the controller when faced with uncertainties unseen during training, such as external disturbances; and (e) **Scalability** of the proposed method to high-dimensional complex systems.

We illustrate four simulated robotic modeling and control problems to show the general applicability of DeSKO. First of all, the classic control problem of CartPole balancing from the control and Reinforcement Learning (RL) literature (Barto et al., 1983) is illustrated. Then, we consider more complicated high-dimensional continuous control problems of robots, such as the legged robot HalfCheetah and the soft robotic arm SoPrA. We simulate the HalfCheetah in the MuJoCo physics engine (Todorov et al., 2012) and the SoPrA (Toshimitsu et al., 2021) in the DRAKE simulation toolbox (Tedrake & the Drake Development Team, 2019). Lastly, we apply DeSKO to autonomous systems in cell biology, i.e., biological gene regulatory networks (GRN) (Elowitz & Leibler, 2000). To further investigate how does DeSKO perform when faced with uncertainty, we introduce process and observation noise to the CartPole and GRN examples, thus four additional variants of the nominal systems are included as benchmarks. The environments are detailed in the Appendix. We compare

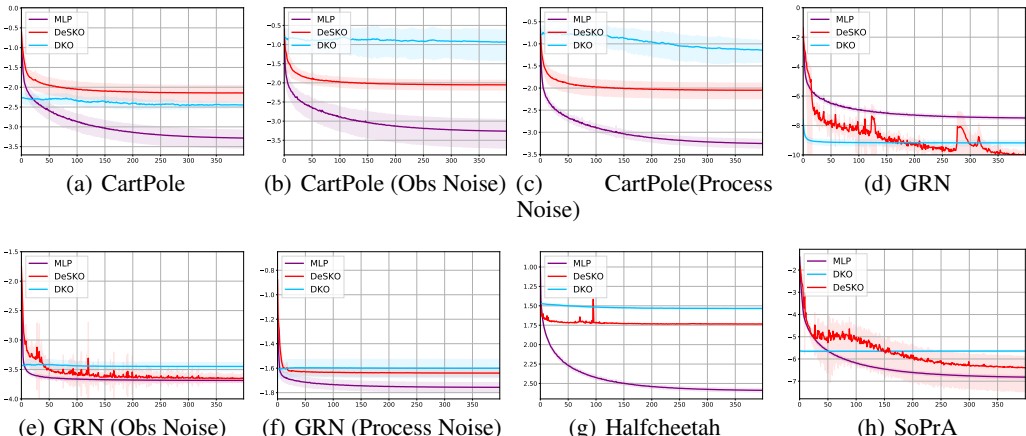

Figure 2: Cumulative prediction error on the validation set. The Y-axis indicates the cumulative mean-squared prediction error in log space over 16 time-steps and the X-axis indicates the training time steps. The shadowed region shows the confidence interval (one standard deviation) over 10 random seeds.

the proposed method with three state-of-the-art baseline methods in terms of modeling and control. **(1)** The Deep Koopman Operator (DKO) (Lusch et al., 2018; Han et al., 2020) learns a neural network as the observable function and the corresponding Koopman operator for forward propagation. In combination with LQR and MPC, it achieves better control performance than reinforcement learning methods such as deep deterministic policy gradient (DDPG) (Lillicrap et al., 2019) on OpenAI gym benchmarks (Brockman et al., 2016). **(2)** An ensemble of ten Multilayer Perceptrons (MLP) (Chua et al., 2018) is trained as a baseline for modeling. Each model is a fully connected NN that maps the current state and action to the next state. The models are trained to minimize the cumulative prediction error over a prediction horizon. The uncertainty could be quantified by the range of

predictions of the models. **(3)** Soft actor-critic (SAC) (Haarnoja et al., 2018) is the state-of-the-art model-free reinforcement learning algorithm. Even though the sample complexity of model-free approaches is much higher than model-based ones, they can typically converge to solutions with better performance. SAC updates the controller to minimize cumulative stage costs, thus implicitly optimizes for a stabilizing controller. Both SAC and DKO could deal with action constraints explicitly. DKO and MLP also used the multi-step loss for training with the same horizon as DeSKO.

For each environment, a training set composed of 40000 state-action pairs and a validation set of 4000 state-action pairs were collected. The actions were collected by uniformly sampling over the action space. Both methods were trained to minimize the cumulative prediction error over a time horizon of 16, and at each update step, a batch of 256 data-points was randomly sampled for the gradient-descent update. The same learning rate 0.001 and decay strategy were used for both methods. SAC iteratively interacts with the environments and updates the control policy. For each environment, $1000k$ steps of state-action-reward pairs were collected for training.

## 5.1 MODELING EVALUATION

The cumulative prediction error during training on the validation set is shown in Figure 2. As shown in the plots, DeSKO achieved better or the same prediction performance than DKO. In particular, in all of the noisy systems, DeSKO achieved better performance; it also scaled better to high-dimensional systems such as HalfCheetah. Nonetheless, MLP achieved better prediction error due to its advantage of huge approximation capacity. However, the key drawback of MLP models is the large computational burden in both forward rollout and backward gradients, making it impossible to obtain an analytical optimal controller or action input.

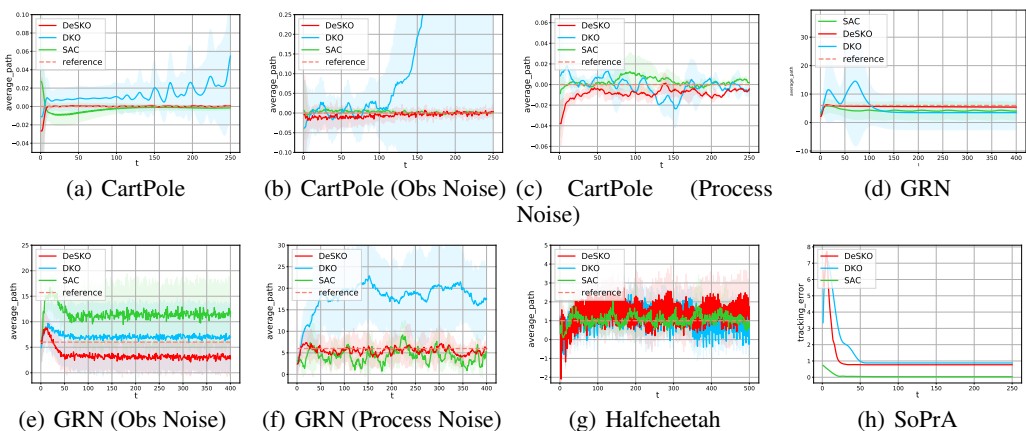

(a) CartPole    (b) CartPole (Obs Noise)    (c) CartPole (Process Noise)    (d) GRN

(e) GRN (Obs Noise)    (f) GRN (Process Noise)    (g) Halfcheetah    (h) SoPrA

Figure 3: State trajectories (a-g) and mean-square tracking error (h). The y-axis indicates the average state trajectory (a-g) or tracking error (h) and the x-axis indicates the time steps in each episode. The shadowed region shows the confidence interval (one standard deviation) over ten random seeds. In (a-g), only the dimensions where a reference signal is tracked are shown.

## 5.2 CONTROL EVALUATION

The control performance of DeSKO and the baselines is compared. A partial state set-point tracking task is assigned to each testing environment. Further details can be found in the Appendix.

To evaluate the performance, we observe the angular position of the pole in CartPole, the concentration of protein 1 in GRN, and the speed in the x-direction in Halfcheetah. For the SoPrA, we record the sequence of mean-square tracking error, because the reference signal is four-dimensional.

As shown in Figure 3, DeSKO attained the best control performance with low average tracking error and variances across different trials in (a-f). In (g) and (h), DeSKO performed comparable to or slightly worse than SAC. In comparison, DKO failed at (a,b,f) and produced large variances in the state trajectories in the GRN systems (d,e).

## 5.3 ROBUSTNESS EVALUATION

It is well-known that optimal controllers designed for learned models could be fragile when faced with unknown impulsive disturbances. Thus, we are also interested in investigating the robustness of the DeSKO controller when faced with unknown external disturbances. To show this, we introduce

periodic external disturbances (every 20 steps) with different magnitudes in the CartPole and GRN systems and observe the performance of each controller. These two environments are chosen as test-beds for robustness due to their higher fragility to disturbances when compared to HalfCheetah and SoPrA. In CartPole, the pole may fall over when interfered by an external force, ending the episode in advance. For this reason, we measure the robustness of controllers with the death-rate, i.e., the probability of falling over after being disturbed. For GRN where the episodes are always of the same length, we measure the robustness of controllers by the variation in the cumulative tracking error. Under each disturbance magnitude, the policies are tested for 100 trials and the performance is shown in Figure 4.

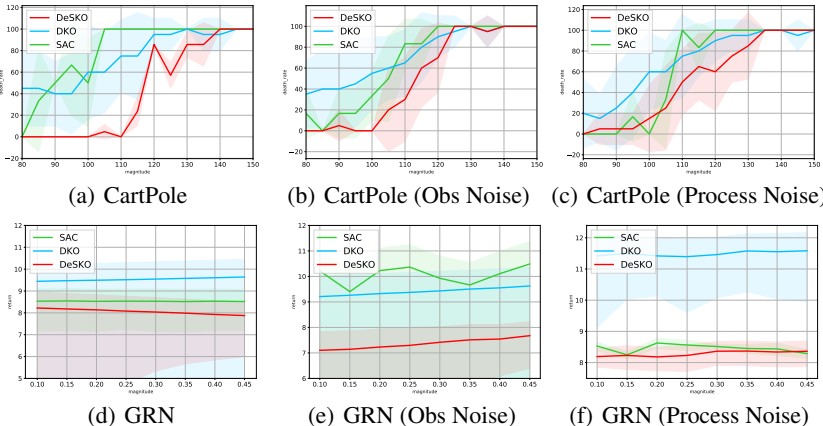

|  |  |  |
|---|---|---|
| (a) CartPole | (b) CartPole (Obs Noise) | (c) CartPole (Process Noise) |
| (d) GRN | (e) GRN (Obs Noise) | (f) GRN (Process Noise) |

Figure 4: Death rate and cumulative tracking error in the presence of persistent disturbances with different magnitudes. The x-axis indicates the magnitude of the applied disturbances. The y-axis indicates the death rate in the CartPole systems (a-c) and the cumulative tracking error in log space in GRN systems (d-e). All of the trained policies are evaluated for 100 trials in each setting.

As shown in Figure 4, DeSKO attained the best robustness in all of the tests. In the CartPole examples, the DeSKO controller can resist disturbances with magnitude up to five times ($100N$) of the maximum control input ($20N$), without any failure (see (a,b)). In (f), the tracking error of DKO is too high to be shown in the plots, and the zoomed-out view of (f) can be found in the Appendix.

### 5.4 ABLATION ON THE ENTROPY CONSTRAINT

We are curious to see what is the effect of the entropy constraint on the robustness of the resulting controller. To show this, we trained three models in CartPole with different minimum entropy thresholds $\mathcal{H}$, and constantly disturb the CartPole to evaluate the robustness of the controller. The death rate of the resulting controllers is shown in Figure 5. As observed in Figure 5, increasing the value of $\mathcal{H}$ helps the resulting controller to be more robust to unknown disturbances, which validates the effect of the entropy constraint.

Figure 5: Ablation study on the Entropy threshold $\mathcal{H}$.

### 6 DISCUSSION AND CONCLUSION

In this paper, we introduced and discussed an efficient model learning approach called the *deep stochastic Koopman operator* (DeSKO). By using deep neural networks to encode the distribution of observables, the DeSKO model is able to infer and propagate the uncertainty in dynamical systems. We developed a robust model predictive control framework based on the learned DeSKO model; the framework guarantees closed-loop stability of the controlled systems. Our experiments show that DeSKO can be applied to high-dimensional complex nonlinear systems, and it outperforms existing deep Koopman operator models and RL algorithms in terms of modeling and control. Our control experiments showed that DeSKO was more robust than the baselines when faced with large external disturbances.

In the future, it could be interesting to investigate the convergence proof of algorithms based on the deep Koopman operator. New Koopman representations that have nonlinear control inputs are also interesting. Another potential future research topic could be control tasks with state constraints.

ACKNOWLEDGMENTS

We thank Lixian Zhang, Miriam Filippi, Lewis Jones, Elvis Nava, Mike Yan Michelis, and Hehui Zheng for their useful comments and insights. DeSKO is supported by the program of China Scholarship Council (No.202006120085). We also want to thank the generous gift from Credit Suisse to the ETH Foundation enabling Soft Robotics Research.

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

# A    THE FROBENIUS-PERRON OPERATOR

Unlike KO, the Frobenius-Perron operator propagates the probabilistic distribution of observables, thus could better capture the uncertainty in the dynamic.

The Frobenius-Perron Operator $\mathcal{P} : \mathcal{L}_1(\mathbb{X}) \rightarrow \mathcal{L}_1(\mathbb{X})$ is able to characterize the evolution of probabilistic distributions (Klus et al., 2016), and it is defined as

$$\int_{\mathbb{A}} \mathcal{P}g(x)\mathrm{d}\mu(x) = \int_{f^{-1}(\mathbb{A})} g(x)\mathrm{d}\mu(x), \forall \mathbb{A} \in \mathcal{B} \tag{14}$$

where $g \in \mathcal{L}_1(\mathbb{X}) := \mathcal{L}_1(\mathbb{X}, \mathcal{B}, \mu)$ is the probabilistic density function (PDF). The measure space $(\mathbb{X}, \mathcal{B}, \mu)$ is composed of the state space $\mathbb{X}$, the $\sigma$-algebra $\mathcal{B}$, and a (probabilistic) measure $\mu$. The FPO $\mathcal{P}$ can be considered a linear, infinite-dimensional representation of the nonlinear, finite-dimensional dynamical system shown in (2) (Klus et al., 2016). Furthermore, the FPO is the adjoint operator of the KO in the following sense (Klus et al., 2016):

$$\langle \psi, \mathcal{P}g \rangle_\mu = \int_{\mathbb{A}} \psi(x)[\mathcal{P}g](x)\mathrm{d}\mu(x) = \int_{\mathbb{A}} [\mathcal{K}\psi](x)g(x)\mathrm{d}\mu(x) = \langle \mathcal{K}\psi, g \rangle_\mu \tag{15}$$

where $\langle \cdot, \cdot \rangle_\mu$ denotes the duality pairing between the $\mathcal{L}_1$ and $\mathcal{L}_\infty$ functions. (15) shows that any probabilistic distribution $g$ can be propagated with the Koopman operator. In this work, we will exploit the above relation to propagate the implicit uncertainty by using the KO.

# B    PROOF OF PROPOSITION 1

*Proof.* The stability proof is composed of two parts, the stability of the nominal system and the stability of the error system between the actual system (9) and the nominal system (5). First, we prove the stability of the nominal system (5). By solving the problem (6)-(7), the optimal control sequence

$$\{c^*_{t|t}, c^*_{t+1|t}, \dots, c^*_{t+H-1|t}\} \tag{16}$$

and the resulting optimal state trajectory

$$\{\hat{\mu}^*_{t|t}, \hat{\mu}^*_{t+1|t}, \dots, \hat{\mu}^*_{t+H-1|t}, \hat{\mu}^*_{t+H|t}\} \tag{17}$$

at instant $t$ are obtained. By appending the control signal produced by the feedback controller $K\hat{\mu}^*_{t+H|t}$ to (16), a suboptimal solution at next time step $t+1$ is given by

$$\{c^*_{t|t}, c^*_{t+1|t}, \dots, c^*_{t+H-1|t}, K\hat{\mu}^*_{t+H|t}\} \tag{18}$$

and

$$\{\hat{\mu}^*_{t|t}, \hat{\mu}^*_{t+1|t}, \dots, \hat{\mu}^*_{t+H-1|t}, \hat{\mu}^*_{t+H|t}, A_K\hat{\mu}^*_{t+H|t}\} \tag{19}$$

where $A_K := A + BK$ denotes the closed-loop transition matrix. Based on this suboptimal solution, we can now prove that the optimal value function $V^*(\hat{\mu}_t)$ is decreasing along the trajectory. Due to the suboptimality of (18) and (19), one has

$$V^*(\hat{\mu}_{t+1}) \leq \sum_{k=1}^{H-1} q(\hat{\mu}^*_{t+k|t}, c^*_{t+k|t}) + q(\hat{\mu}^*_{t+H|t}, K\hat{\mu}^*_{t+H|t}) + p(A_K\hat{\mu}^*_{t+H|t}) \tag{20}$$

$$= V^*(\hat{\mu}_t) + q(\hat{\mu}^*_{t+H|t}, K\hat{\mu}^*_{t+H|t}) + p(A_K\hat{\mu}^*_{t+H|t}) - q(\hat{\mu}^*_{t|t}, c^*_{t|t}) \tag{21}$$

where $q(\hat{\mu}_t, c_t) = \|C\hat{\mu}_t\|^2_Q + \|c_t\|^2_R$ denotes the stage cost and $p(\hat{\mu}_t) = \|C\hat{\mu}_t\|^2_P$ denotes the terminal cost. Thus it follows that

$$V^*(\hat{\mu}_{t+1}) - V^*(\hat{\mu}_t) \leq -q(\hat{\mu}^*_{t|t}, c^*_{t|t})$$

and the optimal value function $V^*(\cdot)$ is a valid Lyapunov function. Therefore the expectation of the nominal state $\mathbb{E}\hat{x}_t = C\hat{\mu}_t$ converges to zero as $t \rightarrow \infty$, that is the nominal state is mean square stable.

Second, let's consider the dynamics of the error system $e_t := \mu_t - \hat{\mu}_t$ defined by the difference of (9) and (5). Substituting the controller (10) into the error system, it follows that

$$e_{t+1} = (A + BK)e_t + g(w_t) \tag{22}$$

Iterate the dynamics of the error system (22) from the initial time instance 1 to $t$ with $e_{t+1} = A_K g(w_t) + A_K^2 g(w_{t-1}) + \cdots + A_K^t g(w_1) + A_K^t e_1$. According to Algorithm 1, the initial instance $e_1$ equals to zero as $\hat{\mu}_1 = \mu_\theta(x_1)$. Then the $L_2$ norm of the error state is given by

$$\|e_{t+1}\| = \|A_K g(w_t) + A_K^2 g(w_{t-1}) + \cdots + A_K^t g(w_1)\|$$
$$\leq \|A_K g(w_t)\| + \|A_K^2 g(w_{t-1})\| + \cdots + \|A_K^t g(w_1)\|$$
$$\leq \beta\|g(w_t)\| + \beta^2\|g(w_{t-1})\| + \cdots + \beta^t\|g(w_1)\|$$

Taking the expectation over the random noise $w_t$, and using the fact that the random noise signal at different time instances are independently distributed, it follows that

$$\mathbb{E}\|e_{t+1}\| \leq \beta\mathbb{E}_{w_t}\|g(w_t)\| + \beta^2\mathbb{E}_{w_{t-1}}\|g(w_{t-1})\| + \cdots + \beta^t\mathbb{E}_{w_1}\|g(w_1)\|$$

According to Assumption (1), we can further infer that

$$\mathbb{E}\|e_{t+1}\| \leq \beta L\mathbb{E}_{w_t}\|w_t\| + \beta^2 L\mathbb{E}_{w_{t-1}}\|w_{t-1}\| + \cdots + \beta^t L\mathbb{E}_{w_1}\|w_1\|$$
$$\leq \beta Lb + \beta^2 Lb + \cdots + \beta^t Lb$$
$$= \frac{(\beta - \beta^t)Lb}{1 - \beta}$$

where the second inequality is a direct result of Assumption 2. As $t \to \infty$, the expectation of the error state norm is bounded by $\frac{\beta Lb}{1-\beta}$. The state of the original system is given by $\mathbb{E}x_t = C(\hat{\mu}_t + \mathbb{E}e_t)$, thus the effect of the error state upon the original state is bounded by $\frac{\beta\sigma Lb}{1-\beta}$, where $\sigma := \|C\|$. Because the nominal state is mean square stable and the error between the actual and nominal state is bounded, the system (2) is proven to be uniformly ultimately bounded. $\qquad\square$

## C    PROOF OF PROPOSITION 2

*Proof.* As the nominal system remains the same as in Proposition 1, the proof for mean square stability of the nominal system is identical as well. We would focus on proving the uniform ultimate boundedness of the error system.

In presence of the approximation residuals, the dynamic of the error system $e_t := \mu_t - \hat{\mu}_t$ is given as follows,

$$e_{t+1} = (A + BK)e_t + g(w_t) + \epsilon_t \tag{23}$$

Iterate the above equation (22) from the initial time instance 1 to $t$, one has

$$e_{t+1} = A_K^t e_1 + \underbrace{A_K \epsilon_t + A_K^2 \epsilon_{t-1} + \cdots + A_K^t \epsilon_1}_{\sum_1^t A_K^k \epsilon_k} + \underbrace{A_K g(w_t) + A_K^2 g(w_{t-1}) + \cdots + A_K^t g(w_1)}_{\sum_1^t A_K^k g(w_k)} \tag{24}$$

. According to Algorithm 1, the initial instance $e_1$ equals to zero as $\hat{\mu}_1 = \mu_\theta(x_1)$. Then the $L_2$ norm of the error state is given by

$$\|e_{t+1}\| = \|\sum_1^t A_K^k \epsilon_k + \sum_1^t A_K^k g(w_k)\|$$
$$\leq \sum_1^t \|A_K^k \epsilon_k\| + \sum_1^t \|A_K^k g(w_k)\|$$
$$\leq \sum_1^t \beta^k \|\epsilon_k\| + \sum_1^t \beta^k \|g(w_k)\|$$

Taking the expectation over the random noise $w_t$, it follows that

$$\mathbb{E}\|e_{t+1}\| \leq \sum_1^t \beta^k \|\epsilon_k\| + \sum_1^t \beta^k \|g(w_k)\|$$
$$\leq \sum_1^t \beta^k \gamma + \sum_1^t \beta^k Lb$$
$$= \frac{(\beta - \beta^t)(Lb + \gamma)}{1 - \beta}$$

where the second inequality is a direct result of Assumption 2 and Assumption 3. As $t \to \infty$, the expectation of the error state norm is bounded by $\frac{\beta Lb + \gamma}{1-\beta}$. Because the nominal state is mean square stable and the error between the actual and nominal state is bounded by a constant $\frac{\beta(Lb+\gamma)}{1-\beta}$, the system (2) is proven to be uniformly ultimately bounded. $\qquad\square$

## D   ROBUST OPTIMAL TRACKING CONTROL

We are concerned with two types of tracking problems, the static set-point tracking problems and the dynamic tracking problems where a time varying reference signal $r$ is given. We will start by showing how to achieve set-point tracking with the proposed controller.

For the set-point tracking problems, the nominal MPC controller is formulated as follows

$$V^*(\hat{\mu}_t) = \min_{c_{0:H-1}} \sum_{k=0}^{H-1} \|C\hat{\mu}_{t+k} - r\|_Q^2 + \|c_{t+k} - u_s\|_R^2 + \|C\hat{\mu}_{t+H} - r\|_P^2 \tag{25}$$
$$\text{s.t. } \hat{\mu}_{t+k+1} = A\hat{\mu}_{t+k} + Bc_{t+k}, c_{t+k} \in \mathbb{U}$$

where $r$ refers to the set-point reference signal and $u_s$ is the set-point control input that sustains the state at the reference. $u_s$ could be obtained by solving the following optimization problem

$$\min_{\hat{\mu}_s, u_s} \|u_s\|^2$$
$$C\hat{\mu}_s = C(A\hat{\mu}_s + Bu_s)$$
$$C\hat{\mu}_s = r$$

where $\hat{\mu}_s$ denotes the unknown nominal set-point observable. Plugging the optimal control input $c_t^*$ of (25) into the controller (10) yields the used control input. The set-point controller is used in the SoPrA arm and GRN examples, where the reference signal is constant.

For dynamic tracking problems like the locomotion of HalfCheetah, the reference signal $r_t$ is time varying and the corresponding MPC is formulated as follows

$$V^*(\hat{\mu}_t) = \min_{c_{0:H-1}} \sum_{k=0}^{H-1} \|C\hat{\mu}_{t+k} - r_{t+k}\|_Q^2 + \|c_{t+k}\|_R^2 + \|C\hat{\mu}_{t+H} - r_{t+H}\|_P^2 \tag{26}$$
$$\text{s.t. } \hat{\mu}_{t+k+1} = A\hat{\mu}_{t+k} + Bc_{t+k}, c_{t+k} \in \mathbb{U}$$

There does not exist a set-point control $u_s$ thus the input regulation is the same as in (6).

## E   EXPERIMENTAL SETUP

The experimental evaluation occured in OpenAI Gym (Brockman et al., 2016) or the Drake simulator (Tedrake & the Drake Development Team, 2019). A snapshot of the adopted environments in this paper can be found in Figure 6.

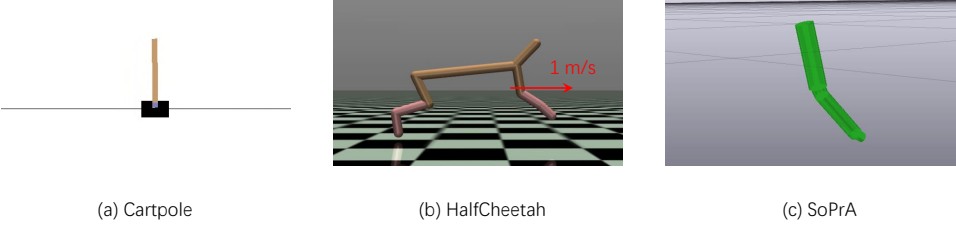

(a) Cartpole                    (b) HalfCheetah                    (c) SoPrA

Figure 6: Snapshot of the environments. The CartPole and HaflCheetah are simulated in OpenAI gym. The SoPrA soft robotic arm is visualized using Drake.

### E.1 CARTPOLE - INVERTED PENDULUM ON A CART

We developed a modified version of CartPole in (Brockman et al., 2016) with a continuous action space instead of a discrete action space. The system contains a horizontally moving cart and has an inverted pendulum attached to it. The cart is fully actuated while the inverted pendulum is unactuated. In this experiment, the controller is expected to maintain the pendulum in its upright, vertical orientation. The action is the horizontal force applied upon the cart ($a \in [-20, 20]$). $x_{\text{threshold}}$ and $\theta_{\text{threshold}}$ represents the maximum of position and angle, respectively, $x_{\text{threshold}} = 10$ and $\theta_{\text{threshold}} = 20°$. The episode ends if $|\theta| > \theta_{\text{threshold}}$ and the episodes end in advance. The episodes for control evaluation are of length 250. For robustness evaluation in Section 5.3, we apply an impulsive disturbance force $F$ on the cart every 20 steps, of which the magnitude ranges from 80 to 150 and the direction is opposite to the direction of control input.

### E.2 HALFCHEETAH - TWO-LEGGED RUNNING ROBOT

HalfCheetah is a legged robot locomotion task adapted from OpenAI Gym (Brockman et al., 2016). The task is to control a two-legged simulated robot to run at the speed of $1 \ m/s$. The control input is the torque applied on each joint, ranging from -1 to 1. The episodes for control evaluation are of length 200.

To achieve dynamic locomotion, a reference signal is first produced for the DeSKO and DKO controllers. In our case, we trained a model-free RL agent using DDPG (Lillicrap et al., 2019) to run forward at the desired speed and record its state trajectory as the reference signal. Nonetheless, this reference signal is suboptimal and could be improved by using model-based planning methods. In the meantime, SAC is trained directly with the reward to run forward at $1m/s$ without the need for a reference signal.

### E.3 SOPRA - SOFT CONTINUUM ROBOTIC ARM

SoPrA is a pneumatic two-segment soft continuum robotic arm (Toshimitsu et al., 2021), built and simulated with the Drake simulation (Tedrake & the Drake Development Team, 2019). The pose of the SoPrA arm is described by just two configuration variables $\phi$ and $\theta$ per segment, which are the relative angle of the plane of bending and the curvature, as is shown in Figure 7. In order to eliminate a singularity in the representation, the following parameterization is adopted

$$\theta_x := \theta \cos(\phi)$$
$$\theta_y := \theta \sin(\phi)$$

and the pose vector $q = [\theta_{x,1}, \theta_{y,1}, \theta_{x,2}, \theta_{y,2}]$, where the subscripts indicate the indexes of the segments. The state is composed of the pose and its derivative, i.e., $x = [q, \dot{q}]$. The controller adjusts the pressure in the six air chambers of SoPrA. Each segment contains three air chambers. Further modeling details can be found in (Toshimitsu et al., 2021). The episodes for control evaluation are of length 250.

### E.4 SYNTHETIC BIOLOGY GENE REGULATORY NETWORKS

The gene regulatory networks (GRNs) considered here are in the nano-scale and their physical properties are vastly different compared to the other examples. Particularly to note is that GRNs can exhibit interesting oscillatory behavior.

In this example, we consider a classical dynamical system in systems/synthetic biology which we use to illustrate the reference tracking task at hand. The GRN is a synthetic three-gene regulatory network where the dynamics of mRNAs and proteins follow an oscillatory behavior (Elowitz & Leibler, 2000). A discrete-time mathematical description of the GRN, which includes both transcription and

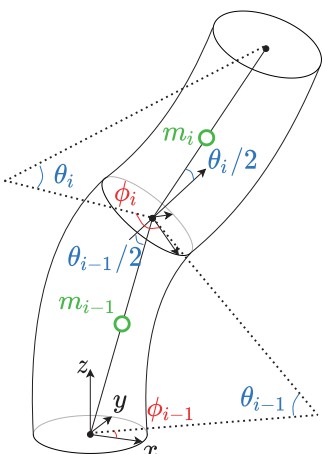

Figure 7: Configuration space of the two-segment soft robotic arm SoPrA, reproduced from Toshimitsu et al. (2021). $\theta_i$ describes the curvature for each bending segment $i$. $\psi_i$ describes the relative angle of the plane of bending for a segment $i$. $m_i$ is the mass of segment $i$.

translation dynamics, is given by the following set of discrete-time equations:

$$
\begin{aligned}
x_1(t+1) &= x_1(t) + dt \cdot \left[ -\gamma_1 x_1(t) + \frac{a_1}{K_1 + x_6^2(t)} + u_1 \right] + \xi_1(t), \\
x_2(t+1) &= x_2(t) + dt \cdot \left[ -\gamma_2 x_2(t) + \frac{a_2}{K_2 + x_4^2(t)} + u_2 \right] + \xi_2(t), \\
x_3(t+1) &= x_3(t) + dt \cdot \left[ -\gamma_3 x_3(t) + \frac{a_3}{K_3 + x_5^2(t)} + u_3 \right] + \xi_3(t), \\
x_4(t+1) &= x_4(t) + dt \cdot \left[ -c_1 x_4(t) + \beta_1 x_1(t) \right] + \xi_4(t), \\
x_5(t+1) &= x_5(t) + dt \cdot \left[ -c_2 x_5(k) + \beta_2 x_2(t) \right] + \xi_5(t), \\
x_6(t+1) &= x_6(t) + dt \cdot \left[ -c_3 x_6(t) + \beta_3 x_3(t) \right] + \xi_6(t).
\end{aligned}
\tag{27}
$$

Here, $x_1, x_2, x_3$ (resp. $x_4, x_5, x_6$) denote the concentrations of the mRNA transcripts (resp. proteins) of genes 1, 2, and 3, respectively. $\xi_i$, $\forall i$ are i.i.d. uniform noise ranging from $[-\delta, \delta]$, i.e., $\xi_i \sim \mathcal{U}(-\delta, \delta)$. During training, $\delta = 0$ and for evaluation $\delta$ is set to 0.5 and 1 respectively in Section 5.3. $a_1, a_2, a_3$ denote the maximum promoter strength for their corresponding gene, $\gamma_1, \gamma_2, \gamma_3$ denote the mRNA degradation rates, $c_1, c_2, c_3$ denote the protein degradation rates, $\beta_1, \beta_2, \beta_3$ denote the protein production rates, and $K_1, K_2, K_3$ are the dissociation constants. The set of equations in Eq.(27) corresponds to a topology where gene 1 is repressed by gene 2, gene 2 is repressed by gene 3, and gene 3 is repressed by gene 1. $dt$ is the discretization time step.

In practice, only the protein concentrations are observed and given as readouts, for instance via fluorescent markers (e.g., green fluorescent protein, GFP or red fluorescent protein, mCherry). The control scheme $u_i$ will be implemented by light control signals which can induce the expression of genes through the activation of their photo-sensitive promoters. To simplify the system dynamics and as it is usually done for the GRN model (Elowitz & Leibler, 2000), we consider the corresponding parameters of the mRNA and protein dynamics for different genes to be equal. More background on mathematical modeling and control of synthetic biology gene regulatory networks can be referred to (Strelkowa & Barahona, 2010; Sootla et al., 2013). In this example, the parameters are as follows:

$$
\forall i: \ K_i = 1, a_i = 1.6, \gamma_i = 0.16, \beta_i = 0.16, c_i = 0.06, dt = 1
$$

In Figure 8, a single snapshot of the state temporal evolution without $\xi$ is depicted. We uniformly initialized between 0 to 5, i.e., $x_i(0) \sim \mathcal{U}(0, 5)$, which is the range we train the policy in Section 5, persistent oscillatory behavior is also exhibiting similar to the snapshot in Figure 8.

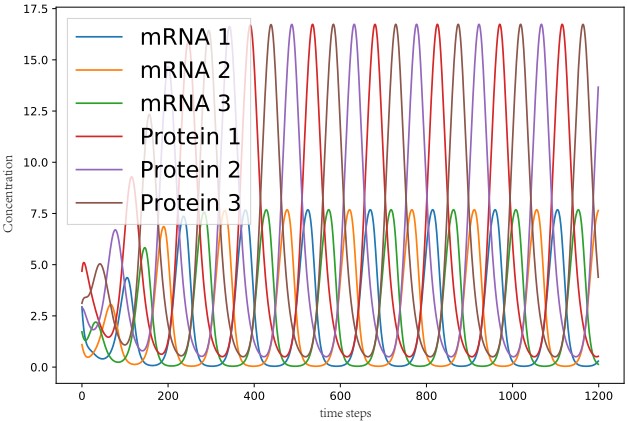

Figure 8: A snapshot of the natural oscillatory behavior of a GRN system consisting of 3 genes. The oscillations have a period of approximately 150 arbitrary time units. The task is to control the concentration of Protein 1 to track a set-point reference signal. The X-axis denotes time and Y-axis denotes the value/concentration of each state.

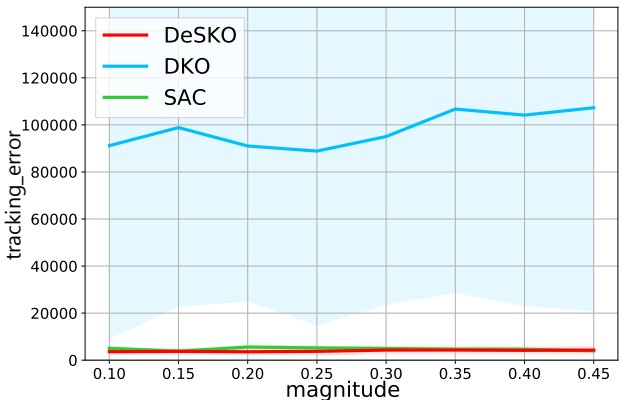

Figure 9: Zoomed-out view of subfigure (f) in Figure 4. As shown above, the tracking error of DKO controller is significantly fragile to disturbances.

## F ADDITIONAL SIMULATION RESULTS

To further show the effectiveness of DeSKO, we also inject noises in the Halfcheetah and SoPrA environments and test the modeling and control performance of DeSKO and the baselines.

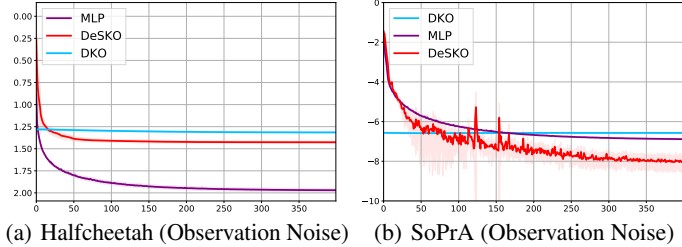

(a) Halfcheetah (Observation Noise)    (b) SoPrA (Observation Noise)

Figure 10: Cumulative prediction error on the validation set. The Y-axis indicates the cumulative mean-squared prediction error in log space over 16 time-steps and the X-axis indicates the training time steps. The shadowed region shows the confidence interval (one standard deviation) over 10 random seeds.

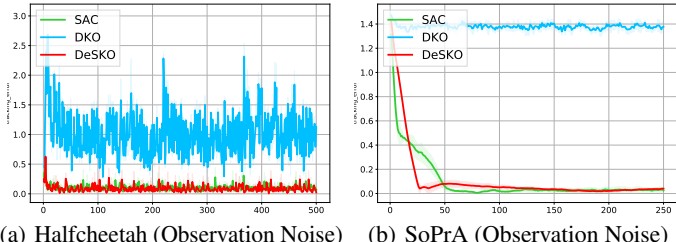

(a) Halfcheetah (Observation Noise)     (b) SoPrA (Observation Noise)

Figure 11: Mean-square tracking error. The y-axis indicates the tracking error and the x-axis indicates the time steps taken during training. The shadowed region shows the confidence interval (one standard deviation) over ten random seeds.

In terms of modelling, DeSKO is still consistently better than DKO, and even outperforms MLP on SoPrA with observation noise. In terms of control, DeSKO also shows better performance than DKO and competitive performance with SAC.

# G  REAL-WORLD EXPERIMENT

To further validate the applicability of DeSKO to real-world control tasks, we tested its control performance on a real SoPrA arm (Toshimitsu et al., 2021), as shown in Figure 12.

The arm was built with silicon and has 6 chambers pneumatically driven by a series of pumps. Each chamber can hold up to 600 millibars of pressure. The state of the arm is characterized by the end position and velocity of each segment, and measured by a motion capture system. The control frequency was set to be 100Hz. We collected a data set of 200000 steps of state-action pairs by randomly actuating the chambers, which took about 40 minutes.

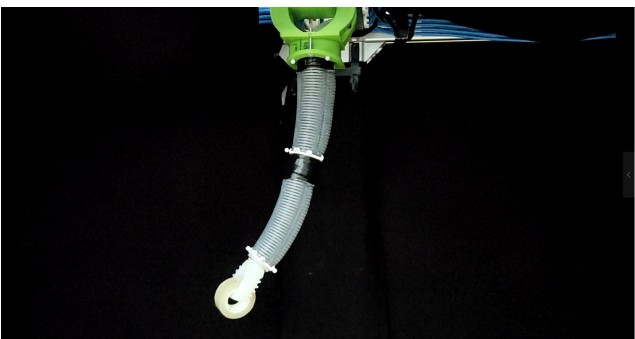

Figure 12: Snapshot of the real SoPrA arm.

## G.1  RESULTS

We train the DeSKO model and the baselines with the same dataset collected in the previous step. Figure 13 shows that both methods can converge during training, and MLP could still achieve the lowest prediction error. Then, we exploit the learned model to design a controller and implement it on the real SoPrA arm on dynamic trajectory tracking problems. The reference trajectories are three letters on the x-y plane (parallel to the ground), and the robot is expected to follow the trajectories with its end-point at the speed of $1cm/s$. The hyperparameters of the controllers are tuned to reach their best performance. As shown in Figure 14, DeSKO outperforms DKO in the tracking accuracy in both scenarios. SAC is not evaluated in the real world, because it requires far more data than the model-based methods and could likely damage the robot during exploring.

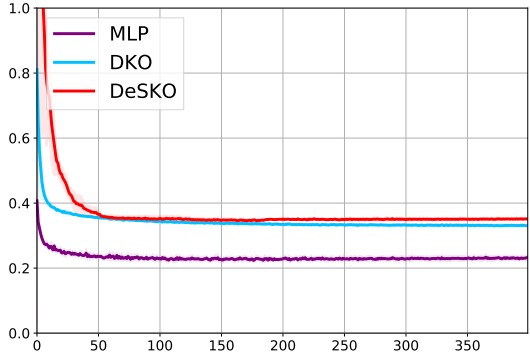

Figure 13: Cumulative prediction error on the validation set. The Y-axis indicates the cumulative mean-squared prediction error in log space over 16 time-steps and the X-axis indicates the training time steps. The shadowed region shows the confidence interval (one standard deviation) over 10 random seeds.

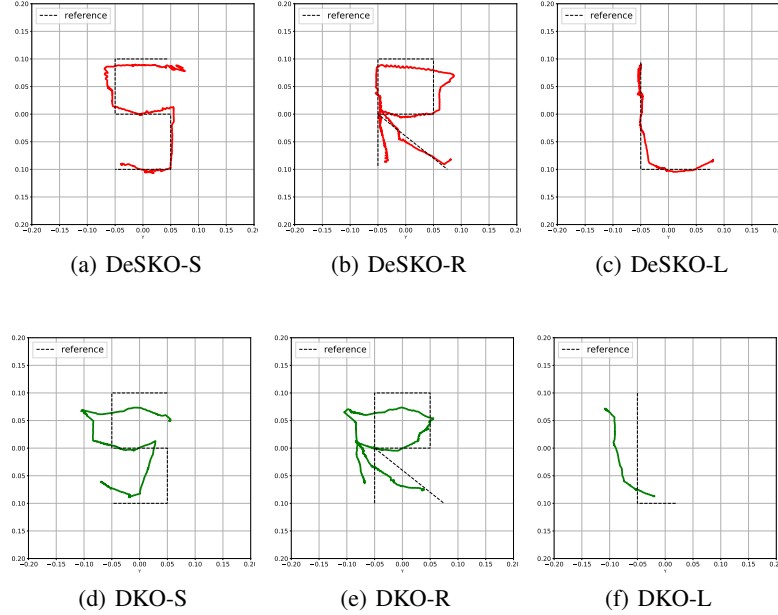

Figure 14: Trajectory tracking with DeSKO and DKO. The above figures show the end-point position of the SoPrA arm on the x-y plane, controlled by the DeSKO and DKO controllers to track the given letter trajectories.

## H HYPERPARAMETERS

The data set is split into the training set composed of 40000 data points, and the validation set composed of 4000 data points. Each data point contains the current state and action input, and the resulting state, i.e. $\{x_t, u_t, x_{t+1}\}$. The data is collected by randomly sampling actions from a uniform distribution over the action space. The DeSKO model is trained with stochastic gradient descent. In our implementation, the ADAM solver (Kingma & Ba, 2017) is used for optimization. At each step during training, a batch of 256 data points is sampled from the training set and used for the model update.

Table 1: Hyperparameters of DeSKO

| Hyperparameters | Value |
|---|---|
| Size of data set $\mathcal{D}$ | 40000 |
| Batch Size | 256 |
| Learning rate | 1e-3 |
| Prediction horzion $H$ | 16 |
| Structure of $\mu_\theta(\cdot)$ | (256,128,64) |
| Structure of $\sigma_\theta(\cdot)$ | (256,128,64) |
| Activation function | ReLU |
| Dimension of observables | 20 |
| Entropy threshold $\mathcal{H}$ | -20 |

