# OpenReview forum: "DeSKO: Stability-Assured Robust Control with a Deep Stochastic Koopman Operator"
_ICLR.cc/2022/Conference — ICLR 2022 Poster_

### Official Review · Reviewer_GCEV · 2021-11-01

**Correctness:** 3
**Technical Novelty And Significance:** 3
**Empirical Novelty And Significance:** 2
**Recommendation:** 6
**Confidence:** 5

**Main Review:**

This work presents a new and elegant approach to modeling uncertainty in data-driven dynamical systems methods. The proposed method is simple and easy to code, yet it achieves impressive results on challenging benchmark control tasks. The guaranteed stability is a nice addition to this framework, which further facilitates the design and application of this approach to new tasks. The extensive evaluation of the approach shows its features in comparison to other methods, and independently. Finally, the versailty of framework and its simplicity opens up new possibilities for future research as partially listed by the authors in the discssion section.

My main concern regarding this work is the somewhat not detailed discussion of related work. Specifically, the following references are missing and should be added:

1. "Learning Koopman invariant subspaces for dynamic mode decomposition", Takeishi et al. 2017
2. "Physics-informed probabilistic learning of linear embeddings of nonlinear dynamics with guaranteed stability", Pan et al. 2020
3. "Forecasting Sequential Data Using Consistent Koopman Autoencoders", Azencot et al. 2020

Moreover, I would like to better understand the differences of the current approach with respect to Morton et al. 2019, and Pan et al. 2020. The former work considers distribution of observables, whereas the latter work has stability guarantees. Similarly, the Azencot et al. 2020 work shows several examples with sensor noise, and they also discuss empirical stability.


minor comments:
	- The reference noa 2016 in Sec. 2 seems to be wrong.
	- Please specify the domain and range of A and B in Eq. (1)
	- Why $\mathcal{K}$ is separable in the dynamics and control?
	- I believe you should change $L_\infty$ to $L_2$ below Eq. (3)
	- I would move Sec. 3.1 & 3.2 to the appendix.
	- The contribution of the Perron-Frobenius to this work is unclear. Please elaborate.
	- The subscript/superscript notation in Eq. (5) is inconsistent with Sec. 3.1
	- The Figs. 2, 4 & 5, and possibly 3 should use a log-scale on the y-axis.
	- Why SAC is not in Fig. 2? Similarly, why MLP is not in Figs. 3 & 4?


**Summary Of The Paper:**

This paper introduces a new control framework that is based on Koopman theory. A key advantage of the proposed approach is the guaranteed stability, yielding a robust closed-loop controller. The method is evaluated on several challenging control benchmarks, and it is compared to several state-of-the-art approaches. The results highlight the advantages of the approach. In particular, in the setting of large external disurbances, the proposed method achieves the best results.

**Summary Of The Review:**

This paper presents an elegant solution to the modeling of uncertainty in dynamical systems for the puprpose of designing closed-loop controllers. The model is backed by control theory and has a guaranteed stability. The method is evaluated on several tasks showing its properties.

---

> ### Author Response · Authors · 2021-11-18
> **Response to major comments**
>
> We greatly appreciate the reviewer’s affirmative comments on the technical contributions of this work and detailed suggestions on the related works and text organization. We have revised the manuscript according to the reviewer’s comments and suggestions. We summarized our revisions of the manuscript in the following to address the reviewer’s concerns.
>
> Takeishi et al. 2017 investigated the data-driven modeling of deterministic autonomous dynamical systems, an approach that can be classified as the “Deep-DMD method”, which has been discussed in Sec.2-Koopman operators of the manuscript. On the other hand, DeSKO can model the stochastic controlled systems by encoding the distribution of observables and assure robust stability of the closed-loop system.
>
> Pan et al. 2020 proposed a Deep-NN based framework with structural parameterization such that the learned Koopman operator is stable. This study is different from DeSKO in two aspects: i) it is focused on learning continuous-time deterministic autonomous systems while DeSKO focuses on stochastic controlled systems; ii) though the dynamic of the observables is stable, the control problem and stability of the original system were not addressed.
>
> Morton et al. 2019 proposed a deep variational Koopman model to encode a distribution of observables and showed its effectiveness in terms of modeling and control. This study is different from DeSKO in two aspects: i) though it also encodes a distribution of observables, this work is focused on deterministic systems with clean datasets, whereas DeSKO is valid for stochastic systems with noisy datasets; ii) DeSKO provides robust closed-loop stability guarantee and is validated on various robotic tasks, while Morton et al. 2019 lacks such guarantees and only showed the results on cartpole and acrobot.
>
> Azencot et al. 2020 proposed a Koopman autoencoder framework and exploited both forward and backward dynamics, and showed its effectiveness in modeling high-dimensional systems. Their work is different from DeSKO in the following aspects: i) their work encourages the model to learn quasi-stable Koopman operator to prevent divergence of prediction, i.e., they are concerned with the stability of the observables, while DeSKO deals with the controlled system and assures the robust stability of the original system, i.e. stability of the states; ii) Azencot et al. 2020 is focused on the learning of autonomous systems, while DeSKO deals with the controlled systems.
>
> The above discussions on the related works are included in the revised manuscript in Section 2.
>
>
> References:
>
> Takeishi N, Kawahara Y, Yairi T. Learning Koopman invariant subspaces for dynamic mode decomposition[C]//Proceedings of the 31st International Conference on Neural Information Processing Systems. 2017: 1130-1140.
>
> Pan S, Duraisamy K. Physics-informed probabilistic learning of linear embeddings of nonlinear dynamics with guaranteed stability[J]. SIAM Journal on Applied Dynamical Systems, 2020, 19(1): 480-509.
>
> Azencot O, Erichson N B, Lin V, et al. Forecasting sequential data using consistent Koopman autoencoders[C]//International Conference on Machine Learning. PMLR, 2020: 475-485.
>
> Proctor J L, Brunton S L, Kutz J N. Generalizing Koopman theory to allow for inputs and control[J]. SIAM Journal on Applied Dynamical Systems, 2018, 17(1): 909-930.
>
> Korda M, Mezić I. Linear predictors for nonlinear dynamical systems: Koopman operator meets model predictive control[J]. Automatica, 2018, 93: 149-160.

---

> > ### Author Response · Authors · 2021-11-18
> > **Response to minor comments**
> >
> > In response to the minor comments:
> >
> > - The reference noa 2016 in Sec. 2 seems to be wrong.
> >
> > Thanks for pointing out the typo, it has been fixed!
> >
> > - Please specify the domain and range of A and B in Eq. (1)
> >
> > Thanks for the reviewer’s advice! The domain has been included in the revised version, the Koopman matrix $A\in \mathbb{R}^{h\times h}$, the control matrix $B\in \mathbb{R}^{h\times m}$, and the observation matrix $C\in\mathbb{R}^{n\times h}$, $h$ is the dimensionality of observables specified by the designer.
> >
> > - Why K is separable in the dynamics and control?
> >
> > Thanks for the reviewer’s question. We separate the Koopman operator into matrices$A$ and $B$ to allow for a control input in the typical structure used in the DMD and EDMD literature. For controlled systems, the observable could take the form of $\psi(x,u)=[\phi(x)^T, u^T]^T$, i.e., taking a linear encoding on $u$ and nonlinear encoding on $x$, respectively. The resulting Koopman matrix would then be a $(n+m)\times (n+m)$ dimensional matrix. Since we are not interested in the prediction of future control input, we can disregard the last $m$ rows of $K$, and separate the remaining parts into two components $A$ and $B$, interacting with the state encoding and action encoding, respectively.
> >
> > - I would move Sec. 3.1 & 3.2 to the appendix; and - The contribution of the Perron-Frobenius to this work is unclear. Please elaborate.
> >
> > We very much appreciate the reviewer’s kind suggestion, we have now moved Sec. 3.2 to the Appendix. Sec. 3.1 introduces the basic idea of Koopman and notations used in this paper, thus we think keeping it in the main text would help the readers better understand the paper.  The aim of Sec 3.2 is to justify the use of Koopman operator from the perspective of its relation to Perron-Frobenius operator. As it plays a less relevant role in the paper, we have moved it to Appendix.
> >
> > - The subscript/superscript notation in Eq. (5) is inconsistent with Sec. 3.1
> >
> > The subscript and superscript notation have been unified, thanks for pointing out!
> >
> > - The Figs. 2, 4 & 5, and possibly 3 should use a log-scale on the y-axis.
> >
> > We have used log-scale in the figures to better compare the performance of different methods in the revised manuscript.
> >
> > - Why SAC is not in Fig. 2? Similarly, why MLP is not in Figs. 3 & 4?
> >
> > Thanks for the reviewer’s question. Fig. 2 shows comparisons in regards to modeling approaches. SAC is a model-free RL method that does not learn the dynamics explicitly, thus SAC can’t be compared against modeling approaches and can therefore not be included in Fig. 2. MLP is an ensemble of multi-layer perceptrons and used as a baseline in terms of modeling, but does not have an associated approach to conduct efficient control tasks, thus is not compared in terms of control nor included in Figs. 3&4.

---

> > > ### Comment · Reviewer_GCEV · 2021-11-29
> > > **response to rebuttal**
> > >
> > > Hi,
> > >
> > > Thank you. The changes made to the manuscript address the concerns I raised in my review. I would be happy to see this work published in its revised form. I prefer to keep my original score.

---

### Official Review · Reviewer_R938 · 2021-11-01

**Correctness:** 3
**Technical Novelty And Significance:** 3
**Empirical Novelty And Significance:** 3
**Recommendation:** 6
**Confidence:** 3

**Main Review:**

Strengths

+ The proposed method is elegant and intuitive, and the resulting algorithm practical and easy to implement.

+ The empirical evaluations are extensive and convincing, with the method performing favorably against strong baselines.

+ The paper is well written and very clear

Weaknesses

- The theoretical results implicitly assume that the learned Koopman embedding (encoded in equation (6)) is exact in the absence of noise across the entire state-space.  This is easily seen by setting $w_t=0$ in equation (10).  Of course, this is never the case in practice, as only prediction error bounds on the parameters $(A,B)$ can be obtained, and these are distribution dependent: i.e., once we switch from the exploratory policy used to identify the parameters (such as random noise, as was used in the experiments) to the closed-loop control policy, we have no guarantees that equations (6) and (10) are valid.  This is one of the main challenges that imitation and reinforcement learning must deal with both practically and theoretically.  Therefore while the main theory result is correct, it is only correct under impractical assumptions and implies nothing about stability of the actual closed-loop system.  Further, the assumption that the lifted system can be written to be linear in the control input (i.e., that the control input enters as $Bu_k$ oin equation (1)) is also very strong, and not supported by the theory, see for example, section 3.2.2 of
@article{korda2018linear,
  title={Linear predictors for nonlinear dynamical systems: Koopman operator meets model predictive control},
  author={Korda, Milan and Mezi{\'c}, Igor},
  journal={Automatica},
  volume={93},
  pages={149--160},
  year={2018},
  publisher={Elsevier}
}



**Summary Of The Paper:**

This paper exploits the duality between the Perron-Frobenius and Koopman operators to formulate a Deep Stochastic Koopman Operator (DeSKO) to control an unknown system.  In particular, in contrast to prior work in this space that assumes a deterministic system, the paper proposes to learn a distribution over observables (encoded via a probabilistic neural network) and linear dynamics that propagate this distribution forward.  Theoretically, it is shown that under certain assumptions, a model-predictive-controller (MPC) consisting of a nominal policy and a stabilizing feedback gain around the nominal trajectory can guarantee robust stability.  Empirically, extensive experiments are conducted across 8 (4 nominal + 4 perturbed) environments, showing that the proposed approach leads to favorable prediction error (Fig. 2), control performance (Fig.3), and robustness (Fig. 4) when compared to other Koopman and Model-Free RL algorithms.  Further an ablation study on the effect of an entropy constraint that is introduced is also performed (Fig. 5).

**Summary Of The Review:**

If the issues with the theoretical statements are adequately addressed, I find this to be a nice paper with convincing empirical results.

---

> ### Author Response · Authors · 2021-11-18
> **Response**
>
> We greatly appreciate Reviewer R938’s insightful question and affirmative comments on the model structure, empirical results and writing. We have revised the manuscript according to the reviewer’s suggestions, and summarized our responses in the following to address the reviewer’s concerns:
>
> The reviewer is making a valid point that Proposition 1 considers only the case where the Koopman operator is exact, and we agree with the reviewer that stability in the case of sub-optimal/approximated Koopman operator needs to be addressed. Therefore, we added as a new theoretical result Proposition 2 to address this concern in Sec. 4.2. We show that the proposed controller (10) can stabilize the system even with an approximated Koopman operator, though the uniform ultimate bound is larger than the bound in the ideal case due to the existence of prediction residuals.
>
> For the reviewer’s concern with the linear control input on latent space, we agree that a more general structure could be proposed where the input is also shifted with a nonlinear function. However, such a Koopman representation relying on an nonlinear control input makes it infeasible to exploit efficient linear control techniques. There is also a bilinear Koopman representation (Goswami et al.(2020), Peitz et al.(2020), Bruder et al.(2021)) that is more general than the linear input representation. However, the bilinear representation can’t cover all types of nonlinear systems and, most importantly, does not preserve the convexity of the resulting control problem either. Due to similar reasons,  Korda et al.(2018)only discussed the generality of bilinear representations and still adopted a linear representation. To recall, the goal of this work is to preserve the convexity of the resulting control problem and leverage linear control techniques to achieve efficient control. Thus, in this paper, we focused on systems that can be approximated by a Koopman representation with linear input and validated its ability to model and control various systems. Designing a more general nonlinear input model that preserves the convexity of the resulting control problem is an interesting research problem, and we would like to leave that for future work.
>
> References:
>
> Korda M, Mezić I. Linear predictors for nonlinear dynamical systems: Koopman operator meets model predictive control[J]. Automatica, 2018, 93: 149-160.
>
> Bruder D, Fu X, Vasudevan R. Advantages of bilinear koopman realizations for the modeling and control of systems with unknown dynamics[J]. IEEE Robotics and Automation Letters, 2021, 6(3): 4369-4376.
>
> Goswami D, Paley D A. Global Bilinearization and Reachability Analysis of Control-Affine Nonlinear[J]. The Koopman Operator in Systems and Control: Concepts, Methodologies, and Applications, 2020, 484: 81.
>
> Peitz S, Otto S E, Rowley C W. Data-driven model predictive control using interpolated Koopman generators[J]. SIAM Journal on Applied Dynamical Systems, 2020, 19(3): 2162-2193.

---

### Official Review · Reviewer_25j7 · 2021-11-02

**Correctness:** 4
**Technical Novelty And Significance:** 2
**Empirical Novelty And Significance:** 3
**Recommendation:** 8
**Confidence:** 4

**Main Review:**

The learning and control of dynamical system from noisy data is not new, but this paper brings a novel angle into solving this problem - stochastic Koopman operator. The learning of the Koopman model and the proposed robust control provides a strong approach in terms of learning accuracy and reliabilily of controller. The proposed approach outperforms existing models and opens up a new perspective for the important problem of learning and control for dynamical systems.



**Summary Of The Paper:**

This paper leverage the Koopman operator theory to represent a nonlinear dynamics as a high dimensional linear dynamics. The map from the state space to the high-dimensional space as well as the Koopman Matrix (linear transition matrix) are learned from data. Similar ideas has been explored in existing works. The contribution of this work is bringing stochasticity into the learning of Koopman models. The paper assumes the data are noisy and try to uncover a Koopman model with noisy data. Another contribution is the joint learning of a control matrix in the Koopman setting and a robust control framework for the proposed method. The proposed robust control framework is based on model predictive control and is provably stable (thanks to the linear transition of the observables). The proposed method is tested on rigid body systems (CartPole and HalfCheetah), a soft robotic arm (SoPrA) and cell biology (GRN). The proposed method outperform existing Koopman-based method (DKO) and RL method (SAC).

**Summary Of The Review:**

Although the building blocks of this paper are not new, the paper leverage these ideas and empirically shows the advantage of the proposed framework. I recommend acceptance of this paper.

---

> ### Author Response · Authors · 2021-11-18
> **Response**
>
> We greatly appreciate the reviewer’s evaluation and affirmative comments on the manuscript! We will continue improving this work with the reviewers.

---

### Official Review · Reviewer_sstK · 2021-11-02

**Correctness:** 3
**Technical Novelty And Significance:** 2
**Empirical Novelty And Significance:** 2
**Recommendation:** 6
**Confidence:** 2

**Main Review:**

The main strengths of the paper:

(1) Extends the Deep-DMD work by allowing uncertainties in the system dynamics.
(2) By projecting the system dynamics to linear spaces, one can explore existing optimal control methods such as LQR for linear systems.
(3) Achieve good control results (i.e. lowest tracking error or and in general more stable) in standard benchmarks.

The  main  weaknesses of the paper includes:

(1) Can use more experiments to show a more convincing picture.

(2) Compared with SAC, the proposed method does not show a clear edge in many cases.

(3) Explanation of how the Perron-Frobenius operator evolves the system distribution is not entirely clear, i.e. why $f^{-1}(A)$ ?

(4) How does equation 3 change the training process compared with the DKO baseline? I assume it has to do with the expected distribution $E_D$ in equation 4 and 5? Please clarify, and also draw the similarity/difference with DKO side by side.

(5) Is multi-step loss also included in DKO training? If not, I don’t think it is a fair comparison. Please clarify.



**Summary Of The Paper:**

In this paper, authors propose  "Deep Stochastic Koopman Operator" which can handle uncertainties in the system dynamics. The authors demonstrate that their approach can produce state loss similar to large capacity MLPs, yet maintains the linear structure in the projected space that is suitable to use linear MPC control methods such as LQR. The proposed method can achieve best tracking performance compared with previous non-stochastic version and SAC trained policies.


**Summary Of The Review:**

The authors proposed a novel method that allows Deep Koopman Operators to handle noisy systems, and demonstrate that their approach beats various baselines. However, there are not enough experiment varieties, and the results are close to the SAC baseline (not by large margin).

---

### Decision · Program_Chairs · 2022-01-20

**Decision:**

Accept (Poster)

**Comment:**

The paper was seen positively by all reviewers. The strength of the paper are:
- Intuitive and interesting combination of Koopman Operators and Optimal Control for Reinforcement Learning
- Convincing experiments on challenging benchmark tasks
- All of the issues of the reviewers (advantages to SAC, gaps in the theory and missing references) have been properly addressed in the rebuttal.

I therefore recommend acceptance of the paper.